# Ketogenic Diet and Vitamin D Metabolism: A Review of Evidence

**DOI:** 10.3390/metabo12121288

**Published:** 2022-12-19

**Authors:** Paraskevi Detopoulou, Sousana K. Papadopoulou, Gavriela Voulgaridou, Vasileios Dedes, Despoina Tsoumana, Aristea Gioxari, George Gerostergios, Maria Detopoulou, George I. Panoutsopoulos

**Affiliations:** 1Department of Clinical Nutrition, General Hospital Korgialenio Benakio, 11526 Athens, Greece; 2Department of Nutritional Science and Dietetics, International Hellenic University, 57400 Thessaloniki, Greece; 3Department of Nutritional Science and Dietetics, Faculty of Health Sciences, University of Peloponnese, 24100 Kalamata, Greece; 4Department of Nutrition and Dietetics, School of Health Science and Education, Harokopio University, 17676 Athens, Greece

**Keywords:** ketogenic diet, vitamin D, weight loss, epilepsy, gene

## Abstract

The ketogenic diet (KD), which is low in carbohydrates and high to normal in fat and protein, has been traditionally used in epilepsy for the last 100 years. More recently, its application in obesity has been introduced. The present review aimed to investigate the effects of the KD on vitamin D. In total, five studies were done in healthy adults, one in subjects with type 2 diabetes, and seven in subjects with epilepsy that assessed the levels of vitamin D pre- and post-intervention. In the majority of studies, increases in circulating vitamin D were reported. The relationship of the KD with vitamin D was explained through several mechanisms: ketone bodies, macronutrient intake, the status levels of other fat-soluble vitamins, weight loss, changes in the hormonal milieu, and effects on gut microbiota. Moreover, potential nutrient–gene-related interactions were discussed. There is still a need to conduct multiple arm studies to compare the effects of the KD versus other diets and better decipher the particular effects of the KD on vitamin D levels and metabolism. Moreover, differentiations of the diet’s effects according to sex and genetic makeup should be investigated to prescribe KDs on a more personalized basis.

## 1. Introduction

Τhe ketogenic diet (KD) has been historically used as a therapeutic diet in subjects with epilepsy [1]. Moreover, it has received extensive interest due to its beneficial effects on several diseases such as type 2 diabetes (T2D), cancer, intestinal disorders, respiratory compromise [2], cardiovascular disorders [3], and obesity [4,5]. Several studies in epileptic children have connected the KD to osteopenia [6]; this relationship was first reported in a study by Hahn et al. 1979 in which children who followed a KD for more than 2 years had low serum vitamin D [7]. Moreover, related dietary patterns such as those high in fat have been connected to altered functions and/or metabolism of vitamin D and other fat-soluble vitamins [8].

Obesity has been related to vitamin D deficiency in epidemiological studies [9]; vitamin D concentration was inversely correlated with fat mass [10]. In parallel, studies in adults prescribing a KD mainly for weight loss mostly led to increases in circulating vitamin D [11].

Moreover, data from human and animal studies suggested that the macronutrient content of a KD [12], the induced increases in ketone bodies [13] and changes in other fat-soluble vitamin status [14], the concurrent hormonal changes [15], alterations in adipose tissue depots [16], and modification of gut microbiome [17,18] may affect vitamin D status and metabolism. Moreover, genes implicated in cholesterol synthesis, hydroxylation, and vitamin D transport can affect vitamin D status and deficiency [19], and potential gene–nutrient interactions may apply [20]. To our knowledge, no previous work has reviewed the exact effects of KDs on vitamin D levels and metabolism nor the potential related underlying mechanisms.

Thus, the aim of the present review was to shed light on the relation of KDs to vitamin D levels and metabolism through an in-depth presentation of intervention studies as well as to clarify potential mechanisms and confounding effects such as those of supplementation. Moreover, potential gene–nutrient interactions are addressed. A clear-cut perspective of existing studies is provided, and potential underlying mechanisms for the observed associations are explained.

## 2. Ketogenic Diets (KDs)

Ketogenic diets, which are low in carbohydrates and high to normal in fat and protein, have been traditionally used in epilepsy for the last 100 years [21].The name “ketogenic” means that they increase ketone body production (acetoacetate, 3-β-hydroxybutyrate, and acetone), which leads to a ketosis state. It is possible that KDs are hypocaloric or supplemented with medium-chain fatty acids to facilitate ketosis [22].

### 2.1. Types and Macro- and Micronutrient Contents of KDs

In general, KDs have a low carbohydrate content that is limited to 5–10% of total kcal daily intake, which corresponds to 10–50 g of carbohydrates [23]. The fat and protein contents of the KD range from 45 to 90% and from 5 to 43%, respectively [23]. The recommended ratio of fat grams to protein-plus-carbohydrate grams in the KD ranges from 4:1 to 2:1 [24]. Several other types of KDs have been introduced: the medium-chain-triglyceride ketogenic diet, the modified Atkins diet, the modified ketogenic diet, the very-low-calorie ketogenic diet (VLCKD), and the ketogenic Mediterranean diet, with the last having the highest (though restricted) carbohydrate content (<30–50 g per day) [23].

The data on the micronutrient content of KDs are scarce. Studies on long-term adoption of KDs point to potential selenium deficits [25,26]; reduced intakes of calcium [27], phosphorus [27], and magnesium [14,27]; as well as a reduced antioxidant capacity [26]. Moreover, increased levels of vitamin E have been documented, while vitamin A decreases after a classic KD and increases after a KD with medium-chain fatty acids [14]. From a clinical perspective, it is recommended that all children with epilepsy under a KD should receive a daily multivitamin supplement as well as calcium and vitamin D (RDA requirements), while supplementations of selenium, magnesium, zinc, phosphorus, iron and copper are optional [1]. However, suboptimal intakes may still be observed despite supplementation [27], and the restrictive nature of a KD can lead to low intakes of phytochemicals, which are not typically included in multivitamins [28].

### 2.2. Uses and Mechanistic Aspects of KDs

Ketone bodies increase during fasting, low carbohydrate intake, and prolonged exercise due to increased fatty acid breakdown and activity of ketogenic enzymes [21,29]. More particularly, ketone production is controlled by the hormone-sensitive enzymes lipase, acetyl-CoA carboxylase, and 3-hydroxy-3-methylglutaryl CoA synthase, which are regulated by insulin, glucagon, and adrenaline [2]. Ketone bodies can be used as fuel for muscles, the heart, and the brain because they can cross the blood–brain barrier [2].

Studies have demonstrated the efficacy of KDs in patients with drug-resistant epilepsy and certain pediatric epilepsy syndromes under medical supervision [30]. This is important in light of evidence that available pharmacological treatments for epilepsy have limited effectiveness even though millions of people suffer from the disease [31]. The potential effectiveness of KDs against epilepsy is mainly due to the fact that ketone bodies can act as direct anticonvulsants [21]. Moreover, a KD diet increases adenosine, which in turn exerts anticonvulsant effects via the receptor A1R [32] and/or by inhibiting DNA methylation [33].

KD has received additional interest due to its beneficial effects on several other diseases such as obesity, T2D, cancer, intestinal disorders, respiratory compromise [2], and cardiovascular disorders [3]. KDs may by also used by athletes to achieve weight loss, but their effects on the quantity and power of muscles are controversial [34].

Weight loss is associated with insulin reduction and redirecting lipids toward oxidation instead of storage [35]. Several mechanisms have been proposed for the beneficial effect of a VLCKD on body weight that include changes in energy expenditure during weight loss and maintenance [36] and appetite suppression (possibly through reductions in insulin levels) [37]. 

Carbohydrate restriction has the greatest effect on fasting and post-prandial glucoregulation [2]. A meta-analysis of randomized controlled trials showed that a KD was more effective than low-fat diets in improving glycemic, weight, and lipid profiles in obese subjects, especially in those with comorbidities such as diabetes [38].

Moreover, KDs decrease serum glucose, insulin, and insulin-like growth factor 1 (IFG-1) levels, which are key molecules in carcinogenesis [39]. In other words, it has been suggested that KDs may have a protective effect against cancer [39]. Indeed, restricted carbohydrate intake through the lowering of insulin may partially suppress these pathways in cancer cells [40]. Interestingly, a population-based study of 9778 participants by Tsujimoto et al. showed that insulinemia was a significant risk factor for cancer death in people with and without obesity [41]. However, in an animal study the long-term administration of a KD had a protumor effect in the kidney [42].

### 2.3. Potential Side Effects of KDs

The first side effects of a KD include vomiting, headache, hypoglycemia, and metabolic acidosis [21,28]. After the first few days, several side effects may appear (gastrointestinal, hepatic, cardiovascular, renal, dermatological, hematologic, and bone effects) [21]. Long-term adverse effects of KDs include high cholesterol levels, nephrolithiasis, growth retardation, and decreased bone mineral density [28].

## 3. Vitamin D

### 3.1. Sources of Vitamin D

The main source of vitamin D is skin synthesis after exposure to UVB light (vitamin D3). As shown in Figure 1, dietary sources of vitamin D are relatively limited and include animal sources such as cod liver oil, fatty fish, liver, and eggs (vitamin D2) [43]. It was noted that plant foods do not contain vitamin D except for supplemented foods and mushrooms, especially those exposed to UV light [43].

### 3.2. Main Steps in the Metabolism of Vitamin D

The main steps in the metabolism of vitamin D are shown in Figure 2. Briefly, vitamin D is hydroxylated in the liver by 25-hydroxylases to form 25-hydroxyvitamin D3 (25(OH)D), which is the major circulating form of vitamin D [44]. Although several hydroxylases such as CYP27A1, CYP2J2/3, and CYP3A4 have been identified, evidence suggests that CYP2R1 is the 25-hydroxylase that is responsible for physiological vitamin D hydroxylation in humans [44]. It was noted that the rate of hydroxylation may be slower in cases of high vitamin D intake [45]. Then, 25(OH)D (bound to vitamin D binding protein) is transferred to the kidneys.

There, 25(OH)D is further hydroxylated by the 1α-hydroxylase enzyme (CYP27B1 gene) into 1,25-dihydroxy vitamin D [1,25(OH)2D], which is the active form of the vitamin [43]. The kidney is the major organ that expresses CYP27B1 [46], while data regarding the extra-renal expression of the enzyme (skin, gastrointestinal tract, bone, and placenta) are controversial [47]. The kidney can also produce 24,25(OH)2D (inactive molecule) through the action of 25OHD-24 hydroxylase (CYP24A1) [48].

### 3.3. Mechanism of Action of Vitamin D

Vitamin D binds to its nuclear vitamin D receptor (VDR), which in turn binds to vitamin-D-responsive elements and regulates gene expression. VDR forms heterodimers with the retinoid X receptors (RXRs), which translocate to the nucleus and increase DNA binding and transcriptional activity [49] (Figure 2). VDR is distributed in several tissues [43].

### 3.4. Concentration of Circulating Vitamin D

Although there is no consensus on the target circulating vitamin D levels and differentiations may apply according to disease status (i.e., osteoporosis), levels below 10 ng/mL indicate insufficiency, while levels 10–20 ng/mL indicate risk of inadequacy [43]. Values greater than 50–100 or 100–150 ng/mL are related to the possible onset of toxicity [43].

## 4. Human Intervention Studies Relating KDs and Vitamin D

The following search terms were used in combinations with the Boolean operator AND to capture the effects of KDs on vitamin D: #1: Ketogenic diet [tiab] OR Ketogenic diet [MeSh] OR Keto diet [tiab] OR Keto diet [MeSh]; #2: low-carbohydrate [tiab] OR low-carbohydrate [MeSh] #3: vitamin D [tiab] OR vitamin D [MeSh] OR cholecalciferol [tiab] OR cholecalciferol [MeSh] OR ergocalciferol [tiab] OR ergocalciferol [MeSh] OR calcitriol [tiab] OR calcitriol [MeSh] OR calcifediol [tiab] OR calcifediol [MeSh] OR 25(OH)D [tiab] OR 25(OH)D [MeSh]. The PubMed database was used as the data source. 

Regarding the criteria for including studies in the present work, it should be noted that only human intervention studies were included. No limitations concerning the definition of a KD were applied. The exclusion criteria were studies on animal and cellular models, human studies in which no intervention was conducted, and studies that considered dietary schemes other than KDs.

### 4.1. Intervention Studies in Healthy Adults and Patients with T2D

Human intervention studies that investigated the effects of KDs on vitamin D in healthy adults and patients with T2D are shown in Table 1. In total, five studies were done in healthy adults [11,50,51,52] and one in subjects with T2D [53]. Regarding healthy participants, they were mostly obese (in three studies, only obese subjects were included), and the KD was prescribed as part of a weight-loss program [11,50,51,52]. Indeed, in all studies the volunteers were initially started on a VLCKD [11,50,51,52]. The duration of intervention ranged from 3 weeks [11] to 12 months [51], while vitamins other than D [50], a vitamin D supplement [54], or no vitamin D [52] was administered. In some studies, a multivitamin supplement was administered, but it was not clear if it also contained vitamin D [11,51]. We noted that in all cases the weight loss was present and increases in circulating vitamin D were documented [11,50,51,52]. The baseline status of vitamin D was suboptimal (<30 ng/mL). In some studies, additional changes were observed in metabolic indices [11,50], inflammatory markers [51], or hormones [50]. An increase in circulating vitamin D3 was reported for patients with T2D on a KD compared to a control group or patients with T2D not on a KD [53]. We noted that only one study compared the effects of a KD to another diet [51]. In this study, vitamin D was increased only in the KD and not in the group following a Mediterranean diet [51].

### 4.2. Intervention Studies in Patients with Epilepsy

Human studies that investigated the effects of KDs on vitamin D in patients with epilepsy are shown in Table 2. In total, seven studies were conducted on subjects with epilepsy [7,55,56,57,58,59,60]. Most of the participants were children who followed a KD for a long period (12–15 months) for medical purposes [7,55,56,57,58,60]. Only one study was conducted on adults [59]. In children, supplemental vitamin D was administered based on the guidelines for children with epilepsy [1] except in one earlier study in which supplementation was tested against a control group [7]. The exact supplemental dose of vitamin D was not reported in most studies. Circulating vitamin D was increased in the first 3 months of treatment and then stabilized [55,58,60] or decreased [56]. The baseline levels of vitamin D were quite heterogenous and ranged from a high percentage of deficiency [7,55,56] to values closer to normal ones (8% of subjects had an insufficient 25(OH)D level) [58]. Changes in bone mineral density or mineral content were also documented in some [55,57] but not all studies [58]. In the study by Molteberg and co-authors, adults with epilepsy were put on a KD or a standard diet [59]. In the KD group, there was an increase in 25(OH)D but the active 1,25(OH)2D decreased, which suggested an interaction of the KD with hydroxylases [59]. It was noted that patients with epilepsy were also on an antiepileptic treatment, which may independently affect vitamin D status [61] and possibly interfere with diet-related effects.

## 5. Potential Effects of KDs on Vitamin D Levels and Metabolism

As presented above, in studies with healthy adults, vitamin D in patients with T2D and patients with epilepsy was increased following a KD. In this section, several potential mechanisms of the observed relationships are proposed.

### 5.1. KD, Ketone Bodies and Vitamin D

In patients with epilepsy following a KD, the low levels of vitamin D that initially were observed were increased after supplementation [7]. Ketone bodies produced by a KD create an acidic environment in which liver and kidney hydroxylases are inactivated, and thus vitamin D is not converted to its active form [13,62]. Acidosis also decreases the vitamin D binding protein, and thus the amount of circulating active vitamin D may be reduced [13]. Similarly, low levels of vitamin D have been associated with the occurrence of diabetic ketoacidosis [63]. In line with this theory, one study that measured both 25(OH)D and its active form 1,25(OH)2D in adult patients with epilepsy following a KD reported that 25(OH)D was increased and 1,25(OH)2D was decreased, which suggested an effect of the KD on hydroxylases [59]. However, it was noted that 1,25(OH)2D has a short half-life of (~4 h) and may not be a reliable index of vitamin D status [45].

### 5.2. KDs, Macronutrient Intake, and Vitamin D

The data regarding macronutrient intakes and their effects on vitamin D status and metabolism are limited. Subjects on low carbohydrates or KDs tend to consume more high-fat foods (meat, butter, eggs, cream, and cheese) even in double quantities, which can lead to an increased dietary intake of vitamin D and increased levels of circulating vitamin D [12]. Indeed, in a recent observational study, subjects following a low-carbohydrate/high-fat diet had higher levels of 25(OH)D (34.9 ± 15.9 ng/mL) compared to those on an eastern European diet (22.6 ± 12.1 ng/mL) [12]. Moreover, fatty acids can interact with cholecalciferol in intestinal absorption [64] and vitamin D supplementation is more effective when given with high-fat meals [65]. In addition, bile acids, which are increased after fat consumption, have been reported to activate VDR [66]. Fat intake may also affect vitamin D effectiveness related to weight changes. For example, mice that consumed a high-fat diet plus vitamin D had a lower body weight and fat mass as well as an increased expression of uncoupling protein 3 compared to mice that consumed a normal diet [67].

The intake of other macronutrients could also affect key metabolic enzymes of vitamin D, although no data exist for KD. For example, dietary protein restriction (9% vs. 20%) in goats, although not the case in KD, led to significant 1,25(OH)2D reduction and increases in 25(OH)D increments; this was connected to the stimulation of CYP2R1 and a reduction in VDBP expression [68].

### 5.3. KDs, Fat-Soluble Vitamin Status, and Vitamin D

Long-term adherence to a KD may cause changes in circulating fat-soluble vitamins such as A (reduction or increase) and E (reduction) in children with epilepsy [14]. Although they have not been tested in the context of a KD, several interactions of fat-soluble vitamins have been described. For example, vitamin E impairs vitamin D absorption by ~15% in vitro [69], and vitamin A also antagonizes vitamin D absorption [70]. Moreover, vitamin A may affect vitamin D actions via the VDR/RXR heterodimer. More particularly, 9-cis-retinoic acid may induce the recruitment of co-activators by the VDR/RXR heterodimer and thus potentiate the responses triggered by vitamin D [71].

### 5.4. KDs, Weight Loss, and Vitamin D

In the above relationships, the confounding effect of weight loss should be taken into account. Vitamin D as a lipid-soluble vitamin is deposited in the adipose tissue. Obese subjects have lower vitamin D concentrations that are possibly due to volumetric dilution, sequestration into adipose tissue, lower sunlight exposure, and lower vitamin D synthesis in the adipose tissue and liver [9]. Indeed, BMI was inversely related to peak serum vitamin D2 levels after an oral supplement of vitamin D2 load or after UV-B irradiation [72], which implied that obese subjects may need larger doses of vitamin D to maintain serum vitamin D levels [73]. Moreover, in a meta-analysis of randomized and non-randomized clinical trials that targeted weight loss (in several ways; not only with a KD), serum 25(OH)D increased by 3.76 nmol/L (95% CI: 2.38, 5.13 nmol/L; *p* < 0.001) [16]. In line with the above observations during a hypocaloric KD, fat mass was inversely related to serum vitamin D levels (rho = −0.040, *p* < 0.05) [52]. From a physiological perspective, this relationship may represent a homeostatic mechanism according to which adipose tissue is a reservoir of vitamin D that can fuel the body when the production of vitamin D is low, such as during the winter. Moreover, an increased expression of 24-hydroxylase CYP24A1 in adipose tissue was reported after weight loss; this degraded both 25OHD and 1,25(OH)2D, which suggested a higher turnover of both 25OHD and 1,25(OH)2D [74].

In all studies assessing the effects of KDs on vitamin D in healthy subjects, weight loss was present [11,50,51,52], which may have masked the net effects of the KD. To our knowledge, there has only been one study that compared the effects of a KD versus another weight-loss diet (Mediterranean diet) on circulating 25(OH)D [51]. In this study, after weight loss via a VLCKD, the serum 25(OH)D concentrations increased from 18.4 ± 5.9 to 29.3 ± 6.8 ng/mL (*p* < 0.0001); while after the Mediterranean diet, the increase in vitamin D was not significant (17.5 ± 6.1 to 21.3 ± 7.6 ng/mL (*p* = 0.06) [51]. In this context, it seems that the type of diet can play a role in the observed perturbations of circulating vitamin D. In a study that included subjects with epilepsy on a KD (supplemented with vitamin D), 25(OH)D was increased only in individuals with a BMI > 25.8 kg/m^2^ while 1,25(OH)2D was reduced in those with a BMI < 25.8 kg/m^2^, which underscored the role of weight status in vitamin D levels [59].

Interestingly, a bidirectional relationship seems to exist between vitamin D and weight loss; i.e., weight loss may lead to increases in circulating vitamin D [16]. Vitamin D supplementation helped in weight-loss interventions according to a meta-analysis [75].

### 5.5. KDs, Hormonal Milieu, and Vitamin D

The KD improves insulin sensitivity, as recently reviewed [15]. Insulin has been also reported to downregulate fibroblast growth factor 23 (FGF23), which is produced by bone cells and is implicated in renal phosphate and vitamin D metabolism [76]. FGF23 physiologically inhibits α-hydroxylase and decreases the formation of active vitamin D [77]. In other words, increasing insulin sensitivity through a KD could lead to reduced levels of FGF23 and potential increases in hydroxylated vitamin D.

### 5.6. KDs, Gut Microbiota, and Vitamin D

KDs have been proposed to favorably modulate gut microbiota by decreasing Firmicutes and increasing Bacteroidetes and microbial diversity [78]. There is some evidence that probiotics increase circulating vitamin D [17,18] and affect protein levels of vitamin D transporters, thus promoting its absorption [18]. In parallel, the potential use of prebiotics in increasing 7-dehydrocholesterol biosynthesis was suggested through a simulation study [79]. However, a study in patients with epilepsy showed vitamin D decreased the levels of short-chain fatty acids [80]. More studies are needed to support or refute the hypothesis of gut-mediated increases in vitamin D when following a KD. 

## 6. Gene–Diet Interactions

In addition to environmental factors, genetic factors also influence vitamin D levels. Variants near genes that are implicated in cholesterol synthesis, hydroxylation, and vitamin D transport can affect vitamin D status and deficiency [19]. More recently, 35 genes and several SNPs have been associated with vitamin D levels [81]. It is thus possible that the genetic variations in vitamin D status may alter individual responses to KD. The following search terms were used in combinations with the Boolean operator AND to capture the effects of KD on vitamin D and to address potential gene–nutrient interactions: #1: Ketogenic diet [tiab] OR Ketogenic diet [MeSh] OR Keto diet [tiab] OR Keto diet [MeSh]; #2: low-carbohydrate [tiab] OR low-carbohydrate [Mesh]; #3: vitamin D [tiab] OR vitamin D [MeSh] OR cholecalciferol [tiab] OR cholecalciferol [MeSh] OR ergocalciferol [tiab] OR ergocalciferol [MeSh] OR calcitriol [tiab] OR calcitriol [MeSh] ] OR calcifediol [tiab] OR calcifediol [MeSh] OR OR 25(OH)D [tiab] OR 25(OH)D [MeSh]; #4: gene [tiab] OR gene [MeSh], #5: polymorphism [tiab] OR polymorphism [MeSh]. The PubMed database was used. It should be noted that no limitations concerning the definition of a KD were applied.

The evidence regarding the effects of diet in general and KDs in particular on vitamin-D-related genes is scarce. A ketogenic diet in an animal model of autism (BTBR^T + Tf/J^ mouse) seemed to reduce body weight and affect 57 genes involved in the nuclear dimer receptor VDR/RXR activation [20]. Moreover, carbohydrates were recently reported to interact with a vitamin-D-related genetic risk score and be related to higher body fat in Asian women [82], which suggested a novel nutrigenetic interaction (high carbohydrates–high genetic score of vitamin D deficiency–high-body-fat phenotype) [82]. Another study reported that those on a relatively low-carbohydrate diet (≤62%) (not KD) who also had a lower number of metabolic risk alleles (genetic risk score using five common metabolic disease-related genetic variants ≤1) had significantly higher levels of 25(OH)D [83], which indicated that low-carbohydrate diets can interact with genes and affect the vitamin D phenotype.

A recent study examined the relationship of VDR polymorphisms and weight loss in a weight-loss program (not KD) with supplementation of vitamin D [84]; while in another, polymorphisms in VDR were related to obesity [85]. Persons with the rs2228570T polymorphism tended to have higher levels of vitamin D compared with those who had a homozygous C allele (*p* = 0.06) [84]. Moreover, persons with the rs731236 heterozygous (CT) allele tended to lose less weight (*p* = 0.06) [84]. In persons not supplemented with vitamin D (control), polymorphisms in the rs1544410 also affected weight loss (*p* = 0.05) [84]. All of the above studies were not direct but suggested that KD–gene interactions may affect perturbations of vitamin D.

## 7. Methodological Considerations

The presented studies were relatively small and most of them were not placebo-controlled. Most studies considered the serum 25(OH)D concentration, which is the best biomarker of vitamin D status [86]. The concentration of 25(OH)D in adipose tissue was not considered in the present studies but could be useful to determine the levels of the vitamin and its metabolites at the cellular level. In the same direction, the measurement of inactive forms of the vitamin in both urine and serum would shed light on changes in vitamin D metabolism during a KD. 

It was noted that no sex-specific analysis was conducted, which may act as a confounder in the observed fluctuations in vitamin D because sex differences in 25(OH)D have been documented [87]. Moreover, given that vitamin D levels have a seasonal variation, the sampling time in intervention studies may have a role in the measured values if the follow-up takes place in a different season [88]. 

Regarding the studies that included patients with epilepsy, it was found that potential drug interactions of the antiepileptic drugs phenobarbital and phenytoin with vitamin D should be considered [43].

## 8. Conclusions

Intervention studies that prescribe a KD for weight loss result in an increase in circulating vitamin D that is mainly due to changes in fat mass, dietary intake, hormonal profile, macronutrient intake, other fat-soluble vitamin status, and gut microbiota alterations. For type 2 diabetes, only one study was identified that also showed increases in circulating vitamin D. Intervention studies in patients with epilepsy are associated with increases in vitamin D if the subjects are supplemented with the vitamin, while other mechanisms may also be present. There is a need to conduct multiple-arm placebo-controlled studies of various caloric intakes (hypo- and normocaloric) to compare the effects of KDs to those of other diets and better decipher their specific effects. Future systematic reviews and meta-analyses can be conducted that consider different sample populations. More analyses are also needed regarding the duration and the amount of KDs in different populations. Moreover, differentiations of the diet’s effects according to sex and genetic makeup should be investigated to prescribe KDs on a more personalized basis.

## Figures and Tables

**Figure 1 metabolites-12-01288-f001:**
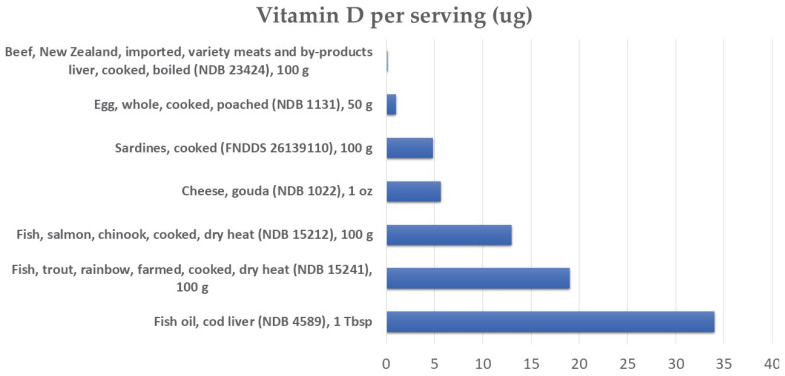
Main dietary sources of vitamin D. NDB: Nutrient Database number corresponding to USDA food database coding: https://fdc.nal.usda.gov (accessed on 27 October 2022).

**Figure 2 metabolites-12-01288-f002:**
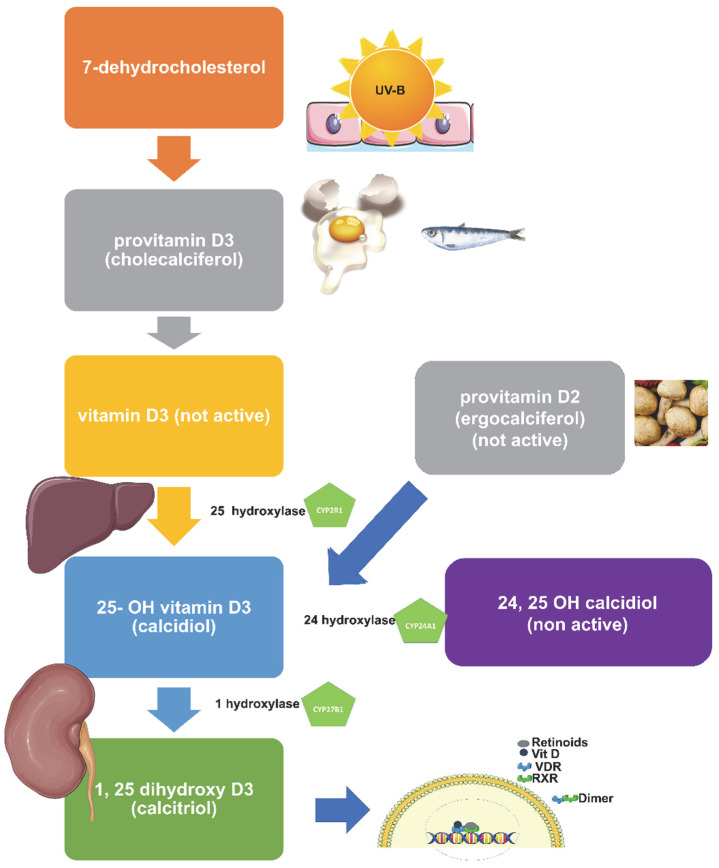
Metabolic steps of vitamin D.

**Table 1 metabolites-12-01288-t001:** Intervention studies with KDs in subjects with overweight, obesity, or T2D diabetes in which vitamin D status was assessed pre- and post-intervention.

Ref	n	Age Range, Mean ± SD (y)	Females/Males	Weight Status	Obesity	Health Status	Study Design	Type of Intervention	Vitamin D Supplement	Duration	Baseline Vitamin D Status	Weight Loss	Effect on Vitamin D	Other Results
	Healthy adults													
Colica, 2017 [11]	40 (20 per group)	18–65 45.4 ± 14.2	Both (% NA)	BMI (Mean ± SD): 30.4 ± 2.6	50% obese	Healthy	Double-blind randomized crossover study; placebo-controlled	VLCKD1: Females: 450–500 kcal, 35–45% PROT, 45–50% fat and 15% CHO(< 20 g). Males: 650–700 kcal, 50–55% PROT, 35–40% FAT, 10% CHO (< 20 g). 50% PROT from synthetic amino acids VLCKD2: Females: 450–500 kcal, 25–35% PROT, 45–50% FAT, 20–25% CHO Males: 650–700 kcal, 45–50% PROT, 35–40% FAT, 15–20% CHO. <10% saturated fat in all diets	Multivitamin, multimineral (not specified)	3 weeks (after a 3 weeks washout)	Serum 25-(OH)2-vitamin D3 Total sample: 21.74 ± 2.38 ng/mL VLCKD1: 21.89 ± 3.88 ng/mL VLCKD2: 22.28 ± 2.69 ng/mL	Yes	↑ 25.7% serum 25-(OH)2-Vitamin D3 with VLCKD2	VLCKD1: ↓ BMI, ↓abdominal fat, ↓ peripheral fat, ↓ HOMA, ↓ glucose, ↓ insulin, ↑ AST, ↑ uric acid, ↑ creatinine VLCKD2: ↓ BMI, ↓abdominal fat, ↓ peripheral fat, ↓ HOMA, ↓ glucose, ↓ insulin
Mongioì 2020 [50]	40	45.8 ± 2.42	40 males	BMI (Mean ± SD): 37.5± 1.1	85% obese 10% overweight	Healthy	Prospective study	CHO <30 g/day Fat 44%, PROT 43% Gradual increase in provided energy VLCKD Phase 1: 600–800 kcal LCKD Phase 2: 800–1000 kcal LCD Phase 3: 1200–1500 kcal Maintenance Phase 4: 1500 and 2000 Kcal	Vitamins (B, C, E), minerals, and omega-3 fatty acids	VLCKD for at least 8 weeks Mean duration 13.5± 0.83 weeks	19.9± 1.1 ng/mL	Yes	↑ 19.9± 1.1 to 38.5± 1.8	↓ Glucose homeostasis ↓ total cholesterol, LDL, TGC, lipids, ↑ HDL-cholesterol, ↓PSA ↑LH, ↑TT No changes in creatinine, uric acid
Perticone 2019 [51]	56 28 in the VLCKD group	46.8 ± 11.0	24 females, 32 males	BMI (Mean ± SD): 39.65 ± 9.7	100% obese	healthy	Clinical trial	VLCKD Phase 1–3: 600–800 kcal (<50 g CHO daily, 10 g of olive oil/day). Phase 4–5 1000–1500 kcal/day	Multivitamin supplement (not specified)	12 months	17.8 ± 5.6 ng/mL 25(OH)D (all) 18.4 ± 5.9 25(OH)D (VLCKD arm)	Yes	↑ 18.4 ± 5.9 to 29.3 ± 6.8 Vitamin D did not increase in the MedDiet group	↓ CRP, ↓ HOMA
dePergola 2020 [54]	22	45 ± 13.9	NA	BMI (Mean ± SD): 31.3 ± 6.2	100% obese or overweight	Healthy	Clinical trial	Low-carbohydrate diet with whey protein 1400–1800 kcal FAT: 50–55%, PROT: 25%, CHO: 15–20% of total calories + nutritional supplement with 18 g of whey proteins (4 g of L-leucine), 4 g of carbohydrates, 1.4 g of lipids, 331 mg of cocoa polyphenols, and several vitamins	5 μg vit D	6 weeks	22.5 (12–26)	Yes	↑ 22.5 (12–26) to 26 (22–35)	↓ Diastolic blood pressure, triglycerides, total cholesterol, pre-albumin, insulin, HOMA, FT3, c-IMT ↑ FMD
Buscemi 2021 [52]	31	18–65 43 ± 11 (intervention)	24 females, 7 males, 20 control group (25% males)	(Mean ± SD): 39.4 ± 6.3	100% obese	Healthy	Placebo-controlled clinical trial	First 20 ± 3 days: VLDKD with industrial meal replacements 600–800 kcal/day, CHO < 50 g; then conventional meals were introduced while maintaining the same nutritional intake	No	10–12 weeks	21.6± 9.9 ng/mL 25(OH)D 29.7 ± 6.7 ng/mL 25(OH)D (control group) Patients with obesity had a higher habitual intake of vitamin D	Yes	↑ 21.6 to 25.8± 10.4 ng/ml	-
	Adults with T2D													
Almseid 2020 [53]	30 patients with T2D on KD, 30 patients with T2D not on KD, 30 controls	30–41	NA	NA	NA	T2D	Case-control study	KD	NA	NA	NA	Yes	↑ Vitamin D3 in patients with T2D on KD (mean ± SE 53.5 ± 0.32) as compared with a control group (mean± SE 57 ± 0.24) and with patients with T2D not on KD (mean ± SE 25.1 ± 1.55)	↑ TT in patients with T2D on KD (mean± SE 427.4 ± 2.52) vs. control group (mean ± SE 422.2 ± 0.24) and patients with T2D not on KD (mean± SE 151.4 ± 1.41); no differences in LDL-cholesterol or HDL-cholesterol

AI: adequate intake; CHO: carbohydrate; c-IMT: carotid intima-media thickness; FMD: flow-mediated dilation; FT3: free triiodothyronine; KD: ketogenic diet; LCKD: low-calorie ketogenic diet; NA: not available or not applicable; PROT: protein; PSA: prostate-specific antigen; PTH: parathyroid hormone; RDA: recommended dietary allowance; SD: standard deviation; SE: standard error; TT: total testosterone; T2D: type 2 diabetes; VLCKD: very-low-calorie ketogenic diet; ↑ increase; ↓ decrease.

**Table 2 metabolites-12-01288-t002:** Intervention studies with KD in patients with epilepsy in which vitamin D status was assessed pre- and post-intervention.

Ref	n	Age Range, Mean ± SD (y)	Females/Males	Weight Status	Health Status	Study Design	Type of Intervention	Vitamin D Supplement	Duration	Baseline Vitamin D Status	Weight Loss	Effect on Vitamin D	Other Results
Children with epilepsy													
Hahn 1979 [7]	5 15 controls	10.4 ± 1.5	3 girls, 2 boys	NA	Patients on anticonvulsant therapy	Placebo-controlled pilot study	Anticonvulsant therapy + KD	Yes	Anticonvulsant drug therapy = 7.4 years; triglyceride ketogenic diet therapy = 2.5 years	14.1 ± 2.5 ng/mL	NA	↑ After supplementation	Decrease in bone mass was observed in the KD group; mean bone mass in the KD + vitamin D group increased by 8.1–0.9% (*p* < 0.001) over 12 months
Bergqvist 2007 [56]	45	5.1 ± 2.7 years	73% (33 33 boys, 27% (12) girls	Weight for age (Z-score) −0.4 ± 1.6 BMI for age (Z-score) −0.3 ± 2.1 (mean± SD)	Epilepsy	Clinical trial	Treatment with the ketogenic diet (KD)	vitamin D (in 14 patients)	15 months	Before KD therapy, 4% had deficient and 51% had insufficient serum 25-OHD levels.	NA	↑ After 3 months and then ↓	↓ PTH
Bergqvist 2008 [55]	25	5–21 7.3 ± 1.9	9 girls, 16 boys	BMI: 16.8 ± 4.4 BMI-for-age z score −0.06 ± 1.6	Epilepsy	Clinical trial	KD 4:1 (g FAT:PROT)	Yes	15 months	54% intake < AI 25-OH D 27.2 ± 13.6 ng/mL 1,25-(OH)2D 25.5 ± 8.3 ng/mL 73% had suboptimal levels (<32 ng/mL)	Yes	↑ In the first 3 months and then stable	↓Whole-body and spine BMC-for-age (0.6 z score/y), ↓ whole-body and spine BMC-for- height (0.7 z score/y and 0.4 z score/y, respectively), ↓ height (0.5 z score/y).
Simm 2017 [57]	29	3.3–17.8 6.4	15 females, 14 males	NA	Epilepsy	Prospective, longitudinal study	PROT:RDA Energy, PROT and FAT:CHO-PROT ratios were adjusted to address weight gain and loss and optimize ketosis	Yes	mean 2.1 years range 0.5–6.5 years	82 nmol/L (range 42–133); 5 patients <50 nmol/L	NA	There were no associations between vitamin D and BMD changes over time	↓ BMD 0.16 SD (relative to age-matched referent children) for every year; ↑ mean urinary calcium/creatinine ratios were elevated (0.77)
Svedlund 2019 [58]	38	6.1 ± 4.8	21 females, 17 males	BMI SDS (median) 0.2 (min-max) 3.3-4.5	Epilepsy, glucose transporter type 1 deficiency syndrome, pyruvate dehydrogenasecomplex deficiency	Prospective longitudinal study	Modified Atkins diet	Yes (14 patients)	24 months	No patient was vitamin D deficient (<12 ng/mL); 8% had an insufficient 25(OH)D level (<20 ng/mL)	No ↑BMI SDS	↑ In the first 6 months and then stable	No effects were observed for bone mass (total body, lumbar spine and hip) or fat mass.
Lee, 2021 [60]	49	0.0–11.7 4.3 ± 3.2	18 girls 31 boys	BMI: 16.4 ± 2.3 Weight SDS: −0.09 ± 1.31 (−3.20–2.95)	Epilepsy	Noncontrolled intervention	KD 3:1 (g fat to nonfat)	Yes D3 (50.8 IU/kg)	12 months	22.4 ± 9.0 42.9% deficiency	NA	↑ In the first 3 months and then stable (not statistically checked)	OR for hypercalciuria was 0.945 (95% confidence interval, 0.912–0.979; *p* = 0.002) per 1.0 ng/mL increment in 25-OH-D3 level.
Adults with epilepsy													
Molteberg 2021 [59]	53	Mean 37.5	33 female 20 male 13 female control 15 male control	BMI (Median): 26.8 (18.7–41.7)	Epilepsy	Placebo- controlled clinical trial	Treatment with a modified Atkins diet, max 16 g of CHO/d (e.g., 5% CHO, 70% FAT, and 25% PROT) Control group: habitual diet, typical Norwegian diet with 43–44%CHO, 34% FAT, 18% PROT	Yes 5–7.5μg	12 weeks	25-OH vit D 60 nmol/l 1,25-OH vit D97 pmol/L	Yes	↑ 25-OH vit D ↓ 1,25-OH2 vit D	↓ PTH, Ca, CTX- 1, P1NP and leptin

AI: Adequate intake; BMC: bone mineral content; BMD: bone mineral density; CHO: carbohydrate; CTX-1: C-terminal telopeptide of type 1 collagen; KD: Ketogenic diet; LCKD: low calorie ketogenic diet; LH: luteinizing hormone; NA: Not available or not applicable; PROT: Protein; PTH: Parathyroid hormone; RDA: Recommended dietary allowance; OR: Odds; SD: standard deviation; SDS: standard deviation score; ↑ increase; ↓ decrease.

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
