# Peer review of "Ketogenic Diet and Vitamin D Metabolism: A Review of Evidence"

_metabolites, 2022, doi:10.3390/metabo12121288_

Round 1

Reviewer 1 Report

1) Title and abstract are appropriate and reflect the manuscript's content. 

2) Introduction is too brief and the rationale for the review was not clearly articulated. 

3) The narrative review part is generally adequate.

4) Fig 1 may not be needed. 

5) Is Fig 2 an original or adapted? 

6) Part 4 - authors described the use of boolean operators but no other details on databases searched and other criteria. Did the authors mean this part to appear like a scoping or systematic review? Also, KD has a wide range of definition; which one was considered by the authors?

7) Table 1 - ref 43 - study design missing

8) Part 6 - again, no mention of databases search & other criteria. 

9) Authors contributions were not updated. 

Author Response

Reviewer #1

1) Title and abstract are appropriate and reflect the manuscript's content.

      We would like to thank the reviewer for his/her comment.

2) Introduction is too brief and the rationale for the review was not clearly articulated.

The introduction has been updated.

3) The narrative review part is generally adequate.

We would like to thank the reviewer for his/her comment.

4) Fig 1 may not be needed.

We agree that the data on vitamin D sources may be known. However, it is good to include such information to realize that animal foods, which are central in the KD, are also good sources of vitamin D.

5) Is Fig 2 an original or adapted?

It is original.

6) Part 4 - authors described the use of boolean operators but no other details on databases searched and other criteria. Did the authors mean this part to appear like a scoping or systematic review? Also, KD has a wide range of definition; which one was considered by the authors?

The authors tried to describe as well as possible the searching algorithm, although the present work was not intended to be a systematic review. The PubMed database was used. All studies which reported results on KD were included. The definition of KD in each study was added in Table1 and 2 to inform the reader. Please see lines 188-193 of the revised manuscript.

7) Table 1 - ref 43 - study design missing

The Table has been updated.

8) Part 6 - again, no mention of databases search & other criteria.

More information was added regarding the searching methodology. Please see lines 366-367 of the updated manuscript.

9) Authors contributions were not updated.

Authors contributions have been added.

Reviewer 2 Report

Dear authors.

Thank you for writing this article.

This is an interesting article reviewing the association between ketogenic diet and vitamin D metabolism. It has a novel topic and the overall writing is of good quality. However, there are some comments that must be answered:

-          In the introduction section lacks some necessary information; any similar previous studies must be cited:  a study by Eastep et al. reported the relationships of high-fat diet and metabolism of lipophilic vitamins (10.15761/IFNM.1000125). All the gaps between your article and the similar previous ones must be brought in this section.

-          Please bold the novelties of your work in the introduction.

-          I suggest adding the pathway of plant-based vitamin D conversion into active vitamin D in figure 2.

-          Studies may also use the term “Keto diet”, except for “ketogenic diet”, so it would be better ad this term to the search strategy.

-          Please write the search strategy completely or bring it in a separate table. The synonyms of vitamin D component is not complete; e.g. colecalciferol, ergocalciferol, calcitriol, and etc. were missed.

-          Although this is not a systematic review, it is better to write which databases or sources were searched.

-          You included a good number of studies; I was wondering why you did not conduct a systematic review and meta-analyses. In terms of different populations, subgroup analyses could be performed.

-          You can add a paragraph of the criteria for including studies to your work.

-          In section 4.1: Reference No. 40 was conducted among obese people, which was reported in details later in the article. So you can either omit this study or bring the details of participants in each study, as well as the details of the intervention such as dose or duration of KD intervention in this section. This would help readers for better judgment.

-          Please add obesity to the characteristics of participants (based on the data of the included studies) in table 2.

-          Please add references in line 365.

-          In the conclusion section, please also write a sentence regarding KD and vitamin D in T2DM populations. Please also suggest future studies to conduct systematic reviews and meta-analyses based considering different sample populations. More analyses also are needed regarding the duration and the amount of KD in different populations.

Author Response

This is an interesting article reviewing the association between ketogenic diet and vitamin D metabolism. It has a novel topic and the overall writing is of good quality. However, there are some comments that must be answered:

-          In the introduction section lacks some necessary information; any similar previous studies must be cited:  a study by Eastep et al. reported the relationships of high-fat diet and metabolism of lipophilic vitamins (10.15761/IFNM.1000125). All the gaps between your article and the similar previous ones must be brought in this section.

We would like to thank the reviewer for his/her comments. The introduction has been accordingly updated.

-          Please bold the novelties of your work in the introduction.

The introduction has been updated.

-          I suggest adding the pathway of plant-based vitamin D conversion into active vitamin D in figure 2.

Figure 2 was updated according to the reviewer’s comment.

-          Studies may also use the term “Keto diet”, except for “ketogenic diet”, so it would be better ad this term to the search strategy.

The term was added. No difference in the deriving studies was observed.

-          Please write the search strategy completely or bring it in a separate table. The synonyms of vitamin D component is not complete; e.g. colecalciferol, ergocalciferol, calcitriol, and etc. were missed.

More search terms have been added.

-          Although this is not a systematic review, it is better to write which databases or sources were searched.

We searched articles published in PUBMED and related bibliography appearing in the references of selected papers. The according methodology has been updated in the manuscript.

-          You included a good number of studies; I was wondering why you did not conduct a systematic review and meta-analyses. In terms of different populations, subgroup analyses could be performed.

The authors tried to describe as well as possible the searching algorithm. However, the present work was not intended to be a systematic review or meta-analysis.

-          You can add a paragraph of the criteria for including studies to your work.

A paragraph regarding criteria for including studies has been added. Please see lines 192-196 of the revised manuscript.

-         In section 4.1: Reference No. 40 was conducted among obese people, which was reported in details later in the article. So you can either omit this study or bring the details of participants in each study, as well as the details of the intervention such as dose or duration of KD intervention in this section. This would help readers for better judgment.

Ref 40 is a review article, so that is why it was not included in Table 1. Data on obesity has been, though, added in Table 2.

-          Please add obesity to the characteristics of participants (based on the data of the included studies) in table 2.

The Table 2 was updated according to the Reviewer’s comment.

-          Please add references in line 365.

The reference has been added.

-        In the conclusion section, please also write a sentence regarding KD and vitamin D in T2DM populations. Please also suggest future studies to conduct systematic reviews and meta-analyses based considering different sample populations. More analyses also are needed regarding the duration and the amount of KD in different populations.

The conclusions were updated.

Round 2

Reviewer 1 Report

I'm fine with the authors' feedback and changes that were done to the paper. 

Reviewer 2 Report

Dear authors, 

Thank you for your responses. The manuscript has now been improved.